# Oxygen Consumption and Basal Metabolic Rate as Markers of Susceptibility to Malignant Hyperthermia and Heat Stroke

**DOI:** 10.3390/cells11162468

**Published:** 2022-08-09

**Authors:** Matteo Serano, Laura Pietrangelo, Cecilia Paolini, Flavia A. Guarnier, Feliciano Protasi

**Affiliations:** 1CAST, Center for Advanced Studies and Technology, University G. d’Annunzio of Chieti-Pescara, 66100 Chieti, Italy; 2DMSI, Department of Medicine and Aging Sciences, University G. d’Annunzio of Chieti-Pescara, 66100 Chieti, Italy; 3DNICS, Department of Neuroscience and Clinical Sciences, University G. d’Annunzio of Chieti-Pescara, 66100 Chieti, Italy; 4Department of General Pathology, Londrina State University, Londrina 86057-970, Brazil

**Keywords:** calsequestrin, excitation–contraction coupling, ryanodine receptor

## Abstract

Calsequestrin 1 (CASQ1) and Ryanodine receptor 1 (RYR1) are two of the main players in excitation–contraction (EC) coupling. CASQ1-knockout mice and mice carrying a mutation in RYR1 (Y522S) linked to human malignant hyperthermia susceptibility (MHS) both suffer lethal hypermetabolic episodes when exposed to halothane (MHS crises) and to environmental heat (heat stroke, HS). The phenotype of Y522S is more severe than that of CASQ1-null mice. As MHS and HS are hypermetabolic responses, we studied the metabolism of adult CASQ1-null and Y522S mice using wild-type (WT) mice as controls. We found that CASQ1-null and Y522S mice have increased food consumption and higher core temperature at rest. By indirect calorimetry, we then verified that CASQ1-null and Y522S mice show an increased oxygen consumption and a lower respiratory quotient (RQ). The accelerated metabolism of CASQ1-null and Y522S mice was also accompanied with a reduction in body fat. Moreover, both mouse models displayed increased oxygen consumption and a higher core temperature during heat stress. The results collected suggest that metabolic rate, oxygen consumption, and body temperature at rest, all more elevated in Y522S than in CASQ1-null mice, could possibly be used as predictors of the level of susceptibility to hyperthermic crises of mice (and possibly humans).

## 1. Introduction

In skeletal muscle, Ca^2+^ release from the sarcoplasmic reticulum (SR) activates muscle contraction and is controlled by a mechanism named excitation–contraction (EC) coupling. Release from the SR is mediated by ryanodine receptors type-1 (RYR1), a large channel placed in the SR terminal cisternae. Mutations in the RYR1 gene are responsible for muscle diseases such as malignant hyperthermia susceptibility (MHS) and central core disease (CCD). MHS is a pharmacogenetic triggered by administration of commonly used volatile anesthetics, i.e., halothane and isoflurane [1,2]. It is now generally accepted that MH crises are caused by a sustained rise in myoplasm Ca^2+^ concentration, as a consequence of an increase in RYR1 opening probability. Common symptoms during crises include skeletal muscle rigidity, rhabdomyolysis, hyperthermia, rise in serum levels of K^+^, and creatine kinase (CK). MH crises may lead to death of patients if the reaction is not promptly interrupted by administration of dantrolene, the only drug approved for acute to treatment of MH patients [3]. MH is genetic disorder inherited in an autosomal dominant pattern: its incidence is estimated to be between 1 in 10,000–30,000 and 1 in 100,000–250,000 depending on geographic location [4,5,6,7]. MH affects all ethnic groups and occurs more frequently in men than in women [5]. Most individuals susceptible to MH have a genetic alteration in the RYR1 gene on chromosome 19 [5]. To date, approximately 70–80% of all MH-susceptible individuals present a mutation in RYR1 gene [5,7]. Heat stroke (HS) is a life-threatening response caused by exposure to a hot/humid climate (environmental HS) or by strenuous physical exercise (exertional HS), which resembles MH episodes in many aspects. Indeed, HS is also characterized by increase in core body temperature above 40 °C (hyperthermia) and rhabdomyolysis of skeletal fibers [8]. The correlation between MHS and HS crises has been hypothesized in several papers [9,10,11,12]. The exposure to hot environments is becoming a serious issue for the human population and for health systems worldwide, being responsible for more deaths than any other natural disasters combined [13,14].

In the past years, several mouse models have been generated with the goal of improving our understanding of the correlation between MHS and HS. One of these mouse models, the heterozygous RYR1^Y522S/WT^ knock-in mouse (abbreviated Y522S), carries a mutation in the RYR1 gene serine in position 522 (corresponding to the human mutation Y524S) that increases the opening probability of the RYR1 channel (i.e., *gain-of-function*). In the homozygous form, the Y522S mutation is embryonically lethal [15], while heterozygous mice are susceptible to lethal hyperthermic episodes when exposed to both halothane or heat. A Ca^2+^-dependent production of reactive oxygen/nitrogen species (ROS/RNS) has been proposed to play a crucial role in the cascade of events leading to uncontrolled SR Ca^2+^ leak, hypercontracture, and, finally, rhabdomyolysis of skeletal fibers [16]. In our laboratory, we also characterized a knockout model lacking skeletal muscle CASQ1 (CASQ1-null mice) that is also MH and HS susceptible [17,18,19]. Although the lack of CASQ1 resulted in a non-lethal phenotype (and no evident abnormalities under standard housing conditions [17]), we soon discovered that CASQ1-null mice exhibit an unexpected increased incidence of spontaneous mortality, and susceptibility to halothane and heat-induced sudden death [18,19]. This phenotype closely resembled that of Y522S mice [16], though less severe, as only homozygotes male mice suffered hyperthermic crises. Later studies supported the idea of impaired Ca^2+^ handling underlying their enhanced anesthetic- and heat-sensitivity: (a) total SR Ca^2+^ store content was greatly reduced [17]; (b) resting Ca^2+^ levels were slightly increased at physiological temperature (37 °C) [18]; and, finally, (c) SR was prone to fast depletion during repetitive stimulation [20].

Since Ca^2+^ is a critical modulator of skeletal muscle metabolism, and because MH and HS crises are hypermetabolic responses, here we assessed the metabolism in basal conditions and under heat stress in CASQ1-null and Y522S knock-in mice.

The results obtained in this study show that hyperthermic mice display increased oxygen consumption rates, basal metabolic rates and heat generation compared to WT. The fact that all parameters are more pronounced in Y522S than in CASQ1-null mice suggest that metabolic rate, oxygen consumption, and body temperature at rest could possibly be used as predictors of the level susceptibility of mice to hyperthermic crises.

## 2. Materials and Methods

### 2.1. Animals

C57bl6 (WT), CASQ1-null, and RYR1^Y522S/WT^ (Y522S) mice were housed in microisolator cages at 20 °C in a 12 h light/dark cycle and provided free access to standard chow and water. The animal experiments carried out in this work were approved by the Italian Ministry of Health (1202/2020-PR). At 4 months of age, mice were euthanized by cervical dislocation, as approved by D. lgs n.26/2014.

### 2.2. In Vivo Experiments

At ~4 months of age, 10 male mice from each group (C57bl6, CASQ1-null, and Y522S) were housed individually in microisolator cages. Animals’ food intake and body weight were registered daily for two weeks, and results were plotted as mean (gr/day) ± standard error of mean (SEM). Then, mice underwent:

-Basal indirect calorimetry. The basal metabolic rate of C57bl6, CASQ1-null, and Y522S mice was registered by indirect calorimetry. Briefly, each mouse was individually placed inside registration chamber of the calorimetry system (Pan Lab/Harvard Instruments, Cornellà, Spain) for 48 h. Measurement obtained during the first 24 h were discarded, because of the adaptation time needed for both mice and system. After adaptation, the volume of carbon dioxide production (VCO_2_) and the volume of oxygen consumption (VO_2_) were registered during the following 24 h using the Oxyletpro calorimeter system for mice (Pan Lab/Harvard Instruments, Cornellà, Spain). VCO_2_ production was then used to estimate O_2_ consumption and other parameters, as previously described [21]. The analysis of the respiratory quotient (RQ) and the energy expenditure (EE) were performed using Metabolism software (Pan Lab/Harvard Instruments, Spain). In detail, they were calculated as follows: RQ = VCO_2_/VO_2_; and EE = (3.815 + (1.32 × RQ)) × VO_2_ × 1.44. After collection, data were interpolated using Prism 9 software (GraphPad Software), to obtain 24 data (one per hour), and, finally, they were plotted as mean curve.-Heat-stress protocol and core temperature recording. To detect changes in metabolic rate and core temperature of mice during heat stress, animals were subjected to the heat-stress protocol during indirect calorimetry experiments. Briefly, mice were placed individually in the registration chambers intended for indirect calorimetry (Pan Lab/Harvard Instruments, Cornellà, Spain) and then they were placed inside an environmental chamber (custom made), where temperature and humidity can be controlled. Considering the different susceptibilities of CASQ1-null and Y522S mice to heat, we designed two different protocols of heat stress in order to avoid lethal crisis [18]. In detail, CASQ1-null mice were subjected to 41 °C for 20 min, while Y522S mice were exposed to 37 °C for 20 min. Two different WT mice groups were used as control, being exposed alternatively to 41 °C or 37 °C, for appropriated comparison with CASQ1-null mice and Y522S, respectively. During heat-stress protocol, indirect calorimetry parameters (see previous paragraph) were registered every 4 min and then plotted as VO_2_ consumption curve. In addition, immediately before and after the protocol, core temperature of each mouse was measured using a rectal thermometer (four channels thermometer TM-946, XS instruments, Milan, Italy). Finally, rise in core temperature during the test was plotted as delta variation.

### 2.3. Body Composition

After euthanasia, specific adipose depots (subcutaneous, epididymal, and retro peritoneal) were anatomically dissected from C57bl6, CASQ1-null, and Y522S mice and weighed. Tissue weights were normalized by dividing the absolute tissue weight by the body weight for each individual mouse and plotted as percentage of body weight.

### 2.4. Western Blot

Extensor digitorum longus (EDL) muscles were rapidly dissected from C57bl6, CASQ1-null, and Y522S mice and quickly frozen in liquid nitrogen until further use. At the appropriate time, muscles were homogenized in RIPA buffer (TRIS HCl 50mM pH 7.4, Triton-X 1%, Deoxycholate 0.25%, NaCl 150 mM, SDS 3%, EDTA 1 mM, Protease inhibitors 2.5%) using a mechanical homogenizer, and then centrifuged for 15 min at 900× *g* at 4 °C. The supernatant was collected, and protein concentrations were determined spectrophotometrically using a BCA quantification kit (ThermoFisher scientific, Waltham, MA, USA). Equal amounts of total protein (30 μg) were resolved in 12% (for uncoupling protein 3, UCP-3), or in 10% (for sarco- or endoplasmic reticulum calcium 1 ATPase, SERCA-1) sodium dodecyl sulfate (SDS) polyacrylamide gel electrophoresis and transferred to nitrocellulose membranes. Blots were then blocked with 5% non-fat dry milk (EuroClone, Pero, Italy) in TBS-T 0.1% for 1 h. Membranes were probed with primary antibody diluted in 5% non-fat dry milk in Tris-buffered saline 0.1% and Tween 20 (TBS-T) overnight at 4 °C, as follows: (a) mouse monoclonal anti-SERCA-1 antibody (1:1000, MA3-912, Invitrogen, Waltham, MA, USA); and (b) rabbit polyclonal UCP-3 antibody (1:1000, PA1-055, Invitrogen, Waltham, USA). Mouse monoclonal anti-glyceraldehyde-3-phosphate dehydrogenase antibody (GAPDH) (1:10,000, TA802519, OriGene Technologies Inc., Rockville, MD, USA) and mouse monoclonal anti-α-actinin sarcomeric (1:5000, A7732, Sigma-Aldrich, St. Louis, MO, USA) were used for loading control. After incubation with primary antibodies, membranes were washed 3 times for 10 min with TBS-T and then incubated with secondary antibodies (1:10,000, horseradish peroxidase–conjugated, Merck Millipore, Burlington, MA, USA) diluted in 5% non-fat dry milk in TBS-T for 1 h at room temperature (RT) and washed in TBS-T. Peroxidase activity was detected using enhanced chemiluminescent liquid (ECL, Perkin-Elmer, Waltham, MA, USA), the bands were visualized using a gel documentation system (UVItec Cambridge Ltd., Cambridge, UK), and the band densitometric quantification of signals was performed using the imaging system Alliance Mini 4 with Alliance 1D MAX software (UVItec Cambridge Ltd., Cambridge, UK).

### 2.5. Cytochrome-c Oxidase Activity

Mitochondria were isolated from dissected EDL muscles to assay cytochrome-c oxidase activity using a colorimetric assay kit (Cytochrome-c Oxidase Assay Kit; Sigma-Aldrich, St. Louis, MO, USA). Samples were placed in an Ultra-Turrax homogenizer (2 × 30 s at 14,500 rpm) in 5 vol of 30 mM KH_2_PO_4_, 5 mM EDTA, 3.0 M sucrose, 0.5 mM dithiothreitol, 0.3 mM phenylsulfonyl fluoride, and 1% (*v/v*), 1 μM leupeptin, 1 μM pepstatin (pH 6.8). All steps for isolation were performed at 4 °C. Mitochondrial fraction was prepared from the total homogenates by differential centrifugation, as previously described [22]. The supernatant from a first homogenate centrifugation (1000× *g* for 10 min) was centrifuged at 14,000× *g* for 35 min. The pellet was then suspended in 30 mM imidazole, 60 mM KCl, and 2 mM MgCl_2_ (pH 7.0) and stored at −80 °C until use. This resuspension was then used to assay cytochrome-c oxidase activity by a colorimetric assay kit (Cytochrome-c Oxidase Assay Kit; Sigma-Aldrich, St. Louise, MO, USA), based on the observation of the decrease in absorbance at 550 nm of ferrocytochrome-c caused by its oxidation to ferricytochrome-c by cytochrome-c oxidase [23]. The results were expressed as cytochrome-c oxidase activity (U/mL × 10^−3^).

### 2.6. Ultrastructural Analyses

EDL muscles were carefully dissected from euthanized C57bl6, CASQ1-null, and Y522S mice, fixed at RT with 3.5% glutaraldehyde in 0.1 M NaCaCO buffer (pH 7.2) and kept at 4 °C in fixative solution until further use for electron microscopy (EM). Fixed muscles were then postfixed in 2% OsO_4_ in the same buffer for 2 h, then block-stained in uranyl acetate replacement. After dehydration, specimens were embedded in an epoxy resin (Epon 812; Electron Microscopy Sciences, Hatfield, PA, USA), as previously described [24,25]. Ultrathin sections (∼50 nm) were cut using a Leica Ultracut R microtome (Leica Microsystem, Wien, Austria) with a Diatome diamond knife (Diatome, Nidau, Switzerland). Longitudinal and transversal sections were viewed in a FP 505 Morgagni Series 268D electron microscope (FEI Company, Brno, Czech Republic) equipped with Megaview III digital camera and Soft Imaging System at 60 kV.

For all quantitative EM analyses, micrographs of non-overlapping regions were randomly collected from transversal sections of internal areas of EDL fibers. Three samples per group (C57bl6, CASQ1-null and Y522S) were analyzed, for a total of 60 muscle fibers for each group. Data were presented as percentages of total values. (A) Mitochondrial and SR volumes were determined using the well-established stereology point-counting technique [26,27], in EM micrographs collected at 8900× or 28,000× magnification, respectively. Briefly, after superimposing an orthogonal array of dots to the electron micrographs, the ratio between numbers of dots falling within mitochondrial or SR profiles and total number of dots covering the whole image was used to calculate the relative fiber volume occupied by mitochondria or SR. (B) In the same set of micrographs used for mitochondrial volume (8900×), the number of severely damaged mitochondria was evaluated and reported as percentage of the total number. Mitochondria with one of the following ultrastructural alterations were classified as severely damaged: (a) presenting disruption of the external membrane, (b) presence of internal vacuolization and/or disrupted internal cristae, and (c) containing myelin figures.

### 2.7. Statistical Analysis

Statistical analysis is reported in the legend of each figure. Data are shown as mean ± SEM. In all cases, differences were considered statistically significant at *p* < 0.05. Chi-squared and one-way and two-way ANOVA tests were performed using Prism 9 software (GraphPad Software, San Diego, CA, USA).

## 3. Results

### 3.1. CASQ1-Null and Y522S Mice Display Increased Metabolic Rate at Rest

To obtain information about body weight and food intake, we monitored the mice for two weeks. No significant differences were found between the amount of chow consumed by CASQ1-null and Y522S mice (respectively, 4.20 ± 0.16 g and 4.22 ± 0.10 g), although both were significantly increased compared to WT (3.34 ± 0.14 g) (Figure 1A).

Interestingly, body weight did not follow the same trend: indeed, hyperthermic mice displayed a lower body weight (CASQ1-null: 26.94 ± 0.48 g; Y522S: 26.35 ± 0.44 g) than WY mice (29.35 ± 0.63 g) (Figure 1B). We then measured the core temperature of the mice at rest and registered a higher core temperature at rest in CASQ1-null and Y522S mice (CASQ1-null: 36.57 ± 0.09 °C and Y522S: 36.44 ± 0.07 °C) compared to WT (36.00 ± 0.16 °C) (Figure 1C). We monitored the same parameters even in younger mice (2 month of age), all being slightly higher in both CASQ1-null and Y522S compared to WT (Appendix A). In addition to food intake, body weight, and core temperature (data in Figure 1), we also estimated the amount of body fat, expressed as percentage of total body weight (Figure 2), after dissecting adipose tissue pads from each animal, as shown in Figure 2A–C. All adipose tissue deposits were then put together, weighed, and data plotted in Figure 2D: body fat mass was significantly higher in WT (6.11 ± 0.15%) than in CASQ1-null (5.62 ± 0.24%) and Y522S (4.78 ± 0.14%) mice.

Mice metabolic rate was evaluated at rest using an indirect calorimetry system (see Methods). Figure 3A indicates that CASQ1-null and Y522S mice have a significantly increased VO_2_ consumption, compared to WT. In CASQ1-null mice, mean VO_2_ consumption was approximately ~25% higher than in WT, while in Y522S, this difference was ~40% greater than in WT during the whole circadian cycle (Figure 3C).

The respiratory quotient (RQ = VCO_2_ produced/VO_2_ consumed), a parameter that gives information about the main metabolic substrate oxidized during cellular respiration (i.e., the relative amount of glucose, lipids, or proteins used to produce energy), was also evaluated. RQ values are different for carbohydrates, lipids, and proteins, because the VCO_2_/VO_2_ ratio is higher in proteins and glucose than in lipids: a ratio between 0.7 and 0.8 indicates the prevalent use of lipids and proteins as substrates, whereas a ratio above 0.8 indicates increasing use of carbohydrates as the main source of energy [21]. Figure 3B shows that WT mice, at rest, had a RQ curve between 0.8 and 0.9 in both light and dark cycles. On the other hand, in CASQ1-null and Y522S mice, RQ had lower values (closer to 0.75), suggestive of a shift toward a more lipid-oriented metabolism. The mean values registered were about ~0.77 in CASQ1-null and ~0.73 in Y522S, while ~0.88 was the average RQ value in WT mice (Figure 3B,D). These results were reinforced by the calculation of basal energy expenditure (BEE), significantly increased in CASQ1-null and in Y522S mice during all circadian cycles, compared to WT: CASQ1-null: 199.3 ± 6.16 kcal/day/kg^0.75^; Y522S: 235.0 ± 5.76 kcal/day/kg^0.75^; WT: 161.0 ± 2.69 kcal/day/kg^0.75^ (Figure 3E).

### 3.2. Increased Oxygen Consumption and Body Temperature during Heat Stress in CASQ1-Null and Y522Smice

After having assessed the metabolic rate of mice at rest (see above), we evaluated oxygen consumption and body temperature during exposure to environmental heat stress. In the past years, we demonstrated that both CASQ1-null and Y522S mice have a high mortality rate during exposure to environmental high temperature, with Y522S mice showing a more severe phenotype [18,28]. In the present work, we modified our heat-stress protocol compared to past publications [18,29,30,31], reducing the length of protocols and reducing temperature for Y522S mice to avoid lethality (Figure 4: see legend). WT mice were used as control in both protocols and exposed to same temperatures (either 37 °C or 41 °C).

We registered VO_2_ consumption during exposure to the heat challenge by indirect calorimetry: both CASQ1-null and Y522S mice displayed a significant increase in VO_2_ consumption along the entire heat-stress protocol compared to control groups (Figure 4A,B). To verify whether the increased metabolism was accompanied by changes in body temperature, we registered core temperature of each mouse before and after heat-stress protocol.

Significant differences were found after heat stress (Figure 4C,D). Core body temperature increased in all groups, even in WT mice:(a)in CASQ1-null mice, the final temperature reached at the end of the stress protocol at 41 °C (was only slightly higher than in control animals): CASQ1-null: 36.49 ± 0.07 °C to 38.93 ± 0.13 °C (vs. WT: 36.0 ± 0.10 °C to 38.49 ± 0.07 °C), with no difference in ΔT (ΔT = +2.44 °C in CASQ1-null vs. ΔT = +2.47 °C in WT). These results seem in disagreement with previous publications showing the susceptibility of CASQ1-null mice to heat stress. However, we should consider that in previous publications the challenge was applied for 1 h instead of only 20 min.(b)in Y522S mice, both final temperature and ΔT were significantly greater at the end of the stress protocol at 37 °C: Y522S: 36.50 ± 0.07 °C to 40.61 ± 0.15 °C (vs. WT: 36.00 ± 0.11 °C to 37.98 ± 0.07 °C), with ΔT in Y522S = +4.11 °C (vs. ΔT in WT = +1.98 °C).

### 3.3. Analysis of Mitochondria and SR Revealed Structural Alterations and Changes in Protein Expression in CASQ1-Null and Y522S Mice

Healthy mitochondria usually exhibit an electron dense dark matrix (Figure 5A–C, black arrows), while when damaged they often appear swollen, with increased volume and a clear matrix (Figure 5B,C, asterisks).

Analysis by EM of mitochondria in EDL muscles indicated that damaged mitochondria were more numerous in CASQ1-null and Y522S than in WT (Figure 5A–C). The visual observations were supported by quantitative analysis: the percentage of mitochondria presenting structural alterations was significantly higher in CASQ1-null and Y522S (respectively, 22.3 ± 0.6% and 31.7 ± 0.4%) than in WT muscles (4.5 ± 0.4%) (Figure 5D). These data confirm previous findings [16,28,32,33]. Furthermore, the relative volume occupied by mitochondria was also significantly increased in both CASQ1-null and Y522S compared to WT mice (CASQ1-null: 6.5 ± 0.3%; Y522S: 6.6 ± 0.2%; WT: 3.9 ± 0.2%) (Figure 5E). Structural alteration of mitochondria could underline dysfunction: indeed, cytochrome-c oxidase activity, a marker of mitochondrial function, was significantly decreased in CASQ1-null and Y522S (0.06 ± 0.01 and 0.04 ± 0.01 U/mL, respectively) compared to WT mice (0.08 ± 0.01 U/mL) (Figure 6A).

In addition, levels of UCP-3 in EDL homogenates, evaluated by WB, were only slightly (+30%) increased in CASQ1-null, while significantly upregulated in Y522S compared to WT (+137%) (Figure 6B,C).

Finally, we quantified the relative fiber volume occupied by SR in EM images of EDL muscle fibers (Figure 7A–C): a significant increase in SR volume was detected in both CASQ1-null and Y522S (respectively, 9.9 ± 0.4% and 8.5 ± 0.5%) compared to WT (6.9 ± 0.4%) (Figure 7D). Additionally, WB analysis of SERCA-1 in EDL homogenates revealed significant overexpression of SERCA-1 in both CASQ1-null (1.10 ± 0.09 A.U.) and Y522S (1.11 ± 0.16 A.U.) muscles compared to WT mice (0.57 ± 0.13 A.U.) (Figure 7E,F).

## 4. Discussion

### 4.1. Main Findings of the Study

Mice carrying the RYR1^Y522S/WT^ mutation have provided, for more than 15 years, a well-accepted model to study a group of human diseases: MHS, HS, and CCD [15,16,28,32]. In addition, our group has also characterized another mouse model, which undergoes hyperthermic crises during exposure to halogenated anesthetics, heat, and exercise: CASQ1-null mice [18,19,29,30,31]. As overheating crises may result from a constitutive hypermetabolic state, in the present work we studied the metabolism of mice in basal conditions and under heat stress, using an indirect calorimetry system. Both Y522S and CASQ1-null mice have reduced weight and body fat but increased food consumption and core temperature (Figure 1 and Figure 2). In addition, indirect calorimetry revealed increased VO_2_ consumption and BEE in resting conditions, but reduced RQ (Figure 3). VO_2_ consumption was increased also under heat stress and accompanied by an excessive rise in core temperature compared to WT (Figure 4). Finally, EM and biochemical analyses revealed: (i) increased mitochondrial and SR volume and increased mitochondrial damage; (ii) increased SERCA-1 and UCP-3 expression (the latter only in Y522S mice) but decreased cytochrome-c activity (Figure 5, Figure 6 and Figure 7).

### 4.2. Hypermetabolism as a Marker of Mice Susceptibility to MH and HS

While Y522S and CASQ1-null mice display a similar phenotype (susceptibility to trigger overheating episodes when exposed to anesthetics, heat, and strenuous exercise [15,16,18,28]), all studies agreed that the phenotype of Y522S mice was more severe than that of CASQ1-nul mice. Indeed, both male and female Y522S mice are susceptible to MH/HS as heterozygous, while homozygous Y522S mice are not viable [15,16]. On the other hand, homozygous CASQ1 knockout are apparently normal under standard housing conditions, with only males that are MH/HS susceptible [18], with crises that need more time to be triggered than on Y522S [29]. The results collected in the present work underline an interesting parallel between the severity of phenotype and several parameters that we studied. For instance, Y522S mice have greater VO_2_ consumption and BEE but reduced RQ and body fat compared to CASQ1-null mice (Figure 1 and Figure 2). A shift of RQ values toward 0.7 in Y522S and CASQ1-null mice indicates that they use more lipids as metabolic fuel, while WT mice use more carbohydrates. As an additional proof-of-concept, higher metabolic rate is also indirectly confirmed by the reduction in body weight and adipose tissue deposits (Figure 1 and Figure 2).

The results collected under stress in the two mouse models (Figure 4) cannot be directly compared, as we used different protocols (41 °C in CASQ1-null vs. 37 °C in YS mice, both for 20 min). However, even if a lower heat-stress temperature was applied to Y522S mice, VO_2_ consumption and core temperature were still higher than in CASQ1-null mice, reinforcing the idea that these parameters are good indicators of the level of hypermetabolism.

### 4.3. Discussing the Role of Mitochondria in the Hypermetabolism of Y522S and CASQ1-Null Mice

MH and HS are hypermetabolic responses to stress: the fact that mitochondrial volume is elevated in the fibers of EDL muscles of both mice (Figure 5, see also [16,32,33]) is alone a possible mechanism leading to hypermetabolism of mice. EDL may be considered representative of the whole-body musculature, because mice are predominantly composed of fast twitch fibers. In addition, it is also clear that the metabolic rate of mitochondria can be also influenced by myoplasmic Ca^2+^ levels, as mitochondrial Ca^2+^ uptake is involved in regulating metabolic pathways (see [34] for review), based on activation of pyruvate, isocitrate, 2-oxoglutarate dehydrogenases [35], and oxidative phosphorylation cascade [36,37].

There is general agreement that in both Y522S and CASQ1-null mice, Ca^2+^ leak from the EC coupling system results in increased levels of cytosolic Ca^2+^ [15,16,17,18,32]. As mitochondria accumulate Ca^2+^ in response to rise in cytosolic Ca^2+^ concentrations [38], possibly via the mitochondrial calcium uniporter (MCU) [34,39,40], an augmented accumulation of mitochondrial Ca^2+^ that can underline increased metabolic rate. An excessive accumulation of Ca in mitochondria has been directly measured in Y522S mice [40].

In both CASQ1-null and Y522S mice, we have also registered an increased mitochondrial damage (Figure 5). Excessive accumulation of Ca^2+^ in mitochondria, has been discussed as a possible mechanism leading to mitochondrial damage [16,32,33]. These results were reinforced by a drastic reduction in cytochrome-c oxidase activity (Figure 5), confirming a general state of mitochondrial injury.

It is well-established in the literature that an increase in mitochondrial activity could result in excessive production of reactive species of oxygen and nitrogen, i.e., ROS and RNS. Increased level of oxidative stress has been proposed as one of the key event underlying MH/HS crises [16,19,29,30,31,33]. Recently, a paper published by Canato and colleagues [41] demonstrated that MCU silencing is followed by a reduction in mitochondrial ROS in Y522S mice fibers. The reduction in ROS in the mitochondria of Y522S fibers also effectively lowered both caffeine-induced Ca^2+^ release and cytosolic Ca^2+^ levels [41]. Our data indicate that oxidative stress is elevated in our mice even in basal conditions (Figure 6), reinforcing the role that ROS may play in MH/HS crises.

In the present study, we also report a significant overexpression of UCP-3 assayed by WB in Y522S compared to WT (Figure 6). Many groups have demonstrated that UCP-3, a member of the uncoupling proteins, reduces oxidative stress through mild uncoupling [36,42,43]. This hypothesis is based on UCPs’ ability to transport protons, allowing for the regulation of mitochondrial membrane potential, dependent on superoxide anion generation from the electron transport chain, generating heat and participating in thermoregulation [44]. Other studies have proposed a direct correlation between UCP-3 and β-oxidation cellular metabolism, attributing a protective role to UCP-3 against triglyceride accumulation [45,46,47]. Taking into consideration these references, we could argue that increased levels of UCP-3 may be in line with both the increased lipid metabolism (i.e., decreased RQ values; Figure 3) and the reduced amount of body fat (Figure 2) in mice that present higher susceptibility to heat. Moreover, UCP-3 overexpression could be also a compensative mechanism of mitochondria to counterbalance increased oxidative stress (Figure 6).

### 4.4. Discussing the Role of SR in the Hypermetabolism of Y522S and CASQ1-Null Mice

Another player that could, in principle, contribute to hypermetabolism and generation of heat is the SERCA pump, which is a protein of the longitudinal SR involved in reuptake of cytosolic Ca^2+^ into the SR stores, hence, in relaxation of muscle fibers [48]. Here, we found: (a) a slight, but significant, increase in SR total volume in both CASQ1-null and Y522S mice and (b) a two-fold increased level of SERCA-1 protein in EDL muscles in both CASQ1-null and Y522S compared to WT (Figure 7). These changes could in principle represent a compensatory mechanism aiming to counterbalance the chronically elevated cytosolic Ca^2+^ concentration caused by the SR Ca^2+^ leak. The SERCA-pump activity requires energy from ATP hydrolysis to reuptake Ca^2+^ against concentration gradient, a process that is known to generate heat. The first clear evidence that SR Ca^2+^ cycling by SERCA can be adapted to generate heat came from studies on deep-sea fishes called blue marlin, in which the heater organ is enriched in SR [49,50]. In line with this, we found increased core temperature in our MH/HS models, not only during heat-stress-related events (Figure 4) but even in basal conditions (Figure 1). Note that increased core temperature is also accompanied with enhanced food intake and decreased adipose tissue (Figure 1 and Figure 2).

## 5. Conclusions

The main message of our study is that CASQ1-null and Y552S mice display an increased metabolism not just in a condition of stress (Figure 4) but also in basal conditions (Figure 3). This intrinsically elevated metabolic rate results in higher body temperature and reduced body weight and fat (Figure 1 and Figure 2). These results may be instrumental to the future development of methods to evaluate the risk of hyperthermic reactions, before animals or individuals are exposed to the stress conditions.

Indeed, the data collected in vivo suggest that there are physiological parameters (such as VO_2_ consumption at rest, basal metabolism, and core temperature) that can be easily measured in living mice and that could, in principle, be used as reliable markers of the level of susceptibility of certain mouse lines to stressors that are known to induce hyperthermic episodes (halogenated, anesthetics, heat, and strenuous exercise).

Our data support the accepted view that accumulation of Ca^2+^ in cytosol and consequently in mitochondria could activate a feed-forward mechanism involving over-production of ROS, which enhance the susceptibility to develop hyperthermic crises. Speculating about the reasons of increased metabolic demand, one could hypothesize that the requirement of additional ATP is dictated by the need to remove the excess of cytosolic Ca^2+^ caused by SR leak. Increased cytosolic Ca^2+^ could act as a dual regulator of muscle metabolism by: (1) forcing enhanced SERCA pump activity, hence creating energy demand and heat generation; and (2) increasing mitochondrial activity, which lead also to a switch of metabolism to lipids.

## Figures and Tables

**Figure 1 cells-11-02468-f001:**
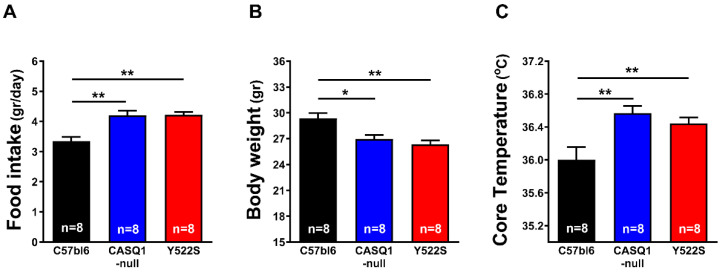
Food intake, body weight, and core temperature. (**A**) Mouse food intake (gr/day); (**B**) mouse body weight (gr); (**C**) core temperature (°C) in basal conditions. Data are shown as mean ± SEM (* *p* < 0.05; ** *p* < 0.01), as evaluated by one-way ANOVA followed by Tukey’s post hoc test. n = number of mice.

**Figure 2 cells-11-02468-f002:**
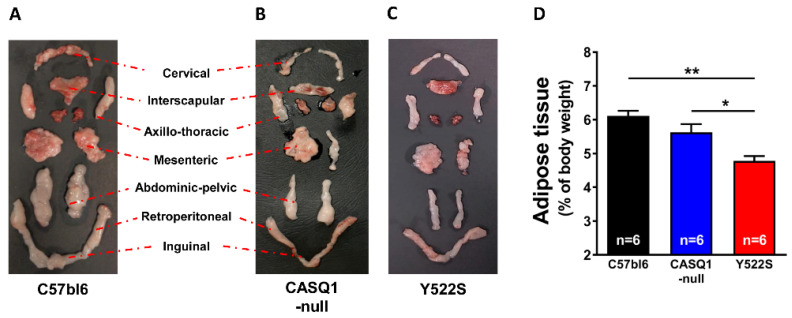
Analysis of adipose tissue. (**A**–**C**) Examples of adipose tissue dissected from C57bl6, CASQ1-null, and Y522S mice. (**D**) Total body fat expressed as percentage of body weight. Data are given as mean ± SEM (* *p* < 0.05; ** *p* < 0.01), as evaluated by one-way ANOVA followed by Tukey’s post hoc test. n = number of mice.

**Figure 3 cells-11-02468-f003:**
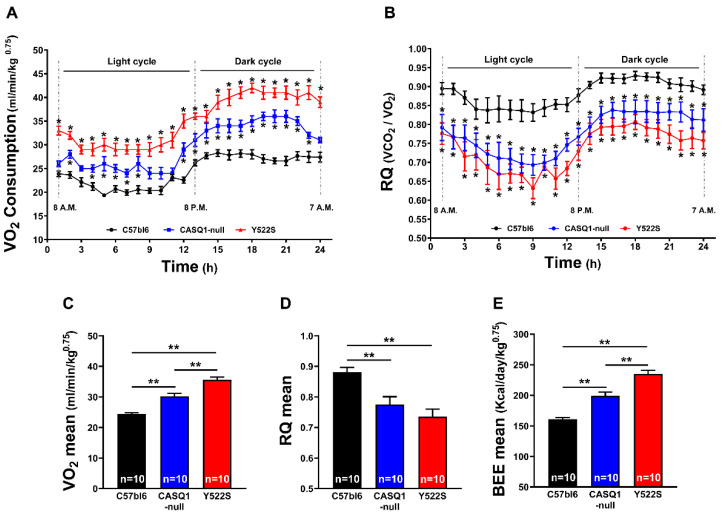
VO_2_ consumption, respiratory quotient (RQ), and basal energy expenditure (BEE) during 24 h. (**A**) Oxygen consumption expressed as mL/min/kg^0.75^. (**B**) Respiratory quotient expressed as VCO_2_/VO_2_ during 24 h. (**C**–**E**) Mean values of O_2_ consumption (panel (**C**)), RQ (panel (**B**)), and basal energy expenditure (panel (**E**)). Legend: light cycle (8 a.m. to 7:59 p.m.); dark cycle (8 p.m. to 7:59 a.m.). Data are shown as mean ± SEM (* *p* < 0.05; ** *p* < 0.01), as evaluated by one-way ANOVA followed by Tukey’s post hoc test. n = number of mice.

**Figure 4 cells-11-02468-f004:**
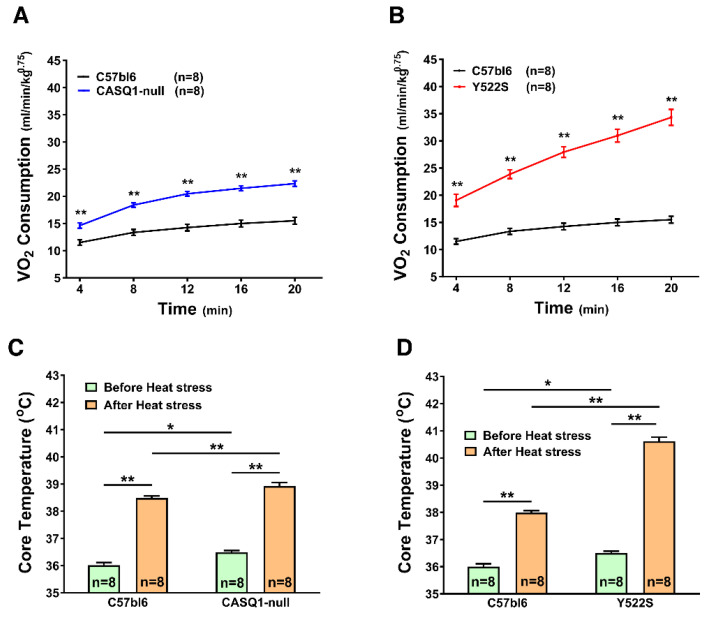
VO_2_ consumption and core temperature during heat stress. (**A**,**B**) Volume of oxygen (VO_2_) consumption (mL/min/kg^0.75^) measured by indirect calorimetry during heat stress. (**C**,**D**) Core temperature before and after the heat-stress protocol. Heat-stress protocol: 20 min at 41 °C for CASQ1-null and WT mice in panels (**A**,**C**); 20 min at 37 °C for Y522S and WT mice in panels (**B**,**D**). Data are given as mean ± SEM (* *p* < 0.05; ** *p* < 0.01), as evaluated by two-way ANOVA followed by Bonferroni’s post hoc test. n = number of mice.

**Figure 5 cells-11-02468-f005:**
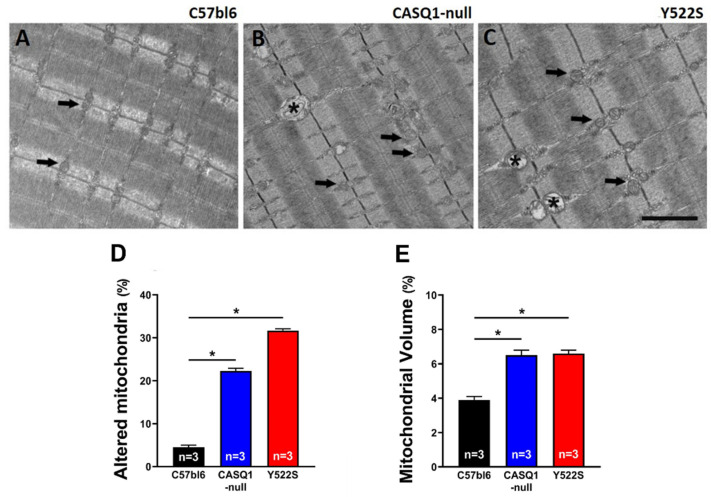
Qualitative and quantitative EM analyses of mitochondria. (**A**–**C**) Representative EM images (from longitudinal sections) of EDL fibers from WT (panel (**A**)), CASQ1-null (panel (**B**)), and Y522S (panel (**C**)) mice. Black arrows point to mitochondria with normal appearance, while asterisks mark altered mitochondria. (**D**,**E**) Quantitative analyses of the percentage of damaged mitochondria (panel (**D**)) and the relative fiber volume occupied by mitochondria (panel (**E**)). Data are given as mean ± SEM (* *p* < 0.05), as evaluated by chi-squared test with Yates’s correction for continuity. Scale bar (**A**–**C**), 1 µm. n = number of mice.

**Figure 6 cells-11-02468-f006:**
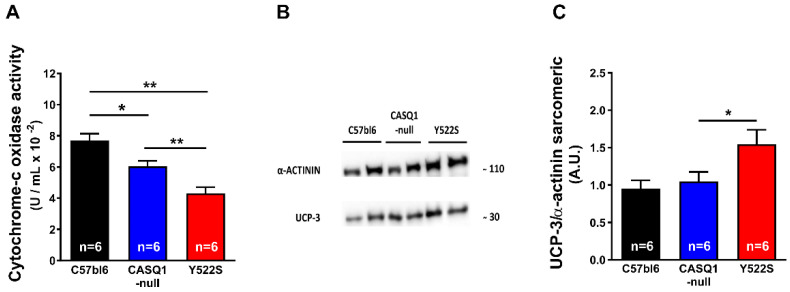
Cytochrome-c oxidase activity and WB analysis of UCP-3. (**A**) Cytochrome-c oxidase activity assayed in EDL homogenates. (**B**,**C**) Representative immunoblots (panel (**B**)) and relative band densities normalized to α-actinin (panel (**C**)) of UCP-3 expression levels in EDL muscle homogenates. Data are given as mean ± SEM (* *p* < 0.05; ** *p* < 0.01), as evaluated by one-way ANOVA followed by Tukey’s post hoc test. n = number of mice.

**Figure 7 cells-11-02468-f007:**
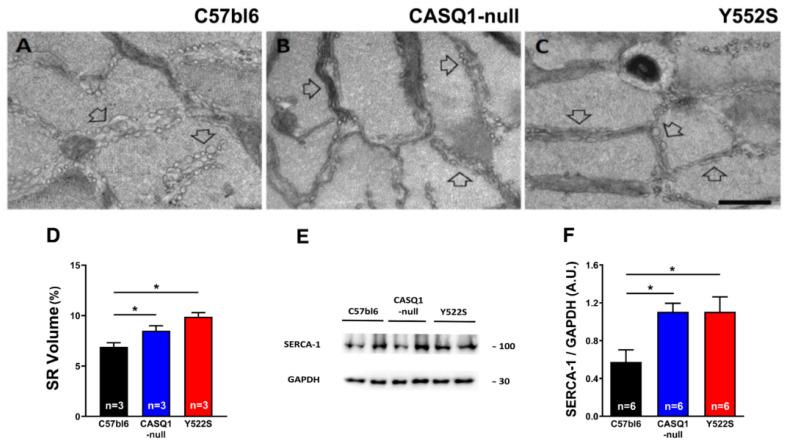
Qualitative and quantitative EM of SR and WB analysis of SERCA-1 expression. (**A**–**C**) Representative EM images of EDL muscle fibers from WT (panel (**A**)), CASQ1-null (panel (**B**)), and Y522S (panel (**C**)) mice showing longitudinal SR cisternae appearance (empty arrows). (**D**) Quantitative analysis of the relative fiber volume occupied by the SR. (**E**,**F**) Representative immunoblots (panel (**E**)) and relative band densities normalized to GAPDH levels of SERCA-1 expression (panel (**F**)) in EDL muscle homogenates. Data are given as mean ± SEM (* *p* < 0.05), as evaluated by chi-squared test with Yates’s correction for continuity (panel (**D**)) or by one-way ANOVA followed by Tukey’s post hoc test (panel (**F**)). Scale bar (**A**–**C**), 1 µm. n = number of mice.

## Data Availability

Not applicable.

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
