# Peer review of "Oxygen Consumption and Basal Metabolic Rate as Markers of Susceptibility to Malignant Hyperthermia and Heat Stroke"

_cells, 2022, doi:10.3390/cells11162468_

Round 1
Reviewer 1 Report
The manuscript sounds relevant and brings deep analysis on metabolic parameters altered in models of malignant hyperthermia and heat stroke, such as increased oxygen consumption and basal metabolic rates, which could be used as predictors of susceptibility to hyperthermic crises. In addition, the manuscript is well written and the approach was very complete, with several techniques to investigate the aims of the study, results very consistent and well-executed discussion, but has some minor issues that need revision and/or correction before publication.
MINOR:
Introduction: Considering the future perspectives of the data obtained by the study, as predictors to hyperthermic crises, I suggest more background on hyperthermic responses in human patients. For example, how is the prevalence of malignant hyperthermia in population?
Lines 200-204: Although saying that the statistical analyses were reported in the figure legends, the item 2.6. describes that Prism 9 software was used to perform the Two-Way ANOVA tests. However, the figure legends mention the using of One-Way ANOVA test. Please, clarify. In addition, the legends of Figures 1, 2, 6 and 7 do not mention the post-hoc test used.
Lines 212-213: Figures 1A and 1B are inverted. A shows the food intake, and B shows the body weight. Please, correct.
Figure 4: If possible, increase the figure to show better the annotations at the graphs.
Lines 288-289: Although no statistical difference between CASQ1-null mice and WT mice considering Dt, the graphs in Figure 4C indicate significant differences. Please, clarify.
Lines 298-299: The Figure 5 does not show “biochemical analyses of cytochrome-c or UCP-3”. Please, correct.
Line 318: Correct to: “Cytochrome-c oxidase activity and WB analysis of UCP-3”.
Reviewer 2 Report
In their submitted work, Serano and collaborators determined at rest and during heat stress the oxygen consumption and energy expenditure in wild type (WT) mice and in two mouse models of Malignant Hyperthermia Susceptibility (MHS). In addition, they analysed mitochondria integrity and functions in EDL muscles. The authors show that mice with MHS are leaner but present a higher food intake and core temperature than WT mice. The phenotype observed is explained by the metabolic switch toward the use of lipids as main source of energy and by mitochondrial dysfunction, even though the latter has been previously reported (doi: 10.1186/s13395-015-0035-9 and 10.1073/pnas.0911496106) and the link between the two observations were not directly addressed in the manuscript. Finally, as the phenotype is more severe in mice bearing the Y522 variant in the Ryr1 gene than in Calsequestrin 1-null mice, the authors proposed that the evaluation at rest of these parameters could be predictive of the severity of MHS.
Even though the results on oxygen consumption and energy expenditure are clear, several major comments need to be addressed before acceptance of the manuscript.
Major comments.
1) Why the study was performed on 4 months old male mice? What about food intake and core temperature in younger mice and/or in female?
2) The authors have previously reported that MHS mice are growth retarded, thus it was expected to observe a lower body weight. Please discuss this point.
3) The fat content was estimated by the adipose tissue mass after dissection. Dexa scan analysis is required for accurate body composition measurement. This is of importance as the figure 2B and C seem not representative of figure 2D. H&E and Perilipin staining will help to determine global architecture and size of the adipocytes, respectively. Is there any difference between white and brown adipose tissue?
4) In figure 5, the authors showed that MHS mice have an increase of the proportion of altered mitochondria. What about the mitochondrial DNA content in muscles?
5) What was the rationale to study UCP3 ? What about AMPK pathway which promotes fatty acid oxidation to protect cells from energy deprivation? What are the levels of UCP3 in muscles with slow- and mix-twitch fibers (i.e. soleus and gastrocnemius, respectively)?
6) ATP consumption on muscle fibers or analysis of the expression of the main players of the mitochondrial respiratory chain, as well as of ROS production, are required to characterise mitochondrial dysfunctions.
7) The representative western blotting used in Figure 6B and 7E are not corresponding to their quantification in Figure 6C and 7F (i.e. huge variability in CASQ1-null mice for UCP3 ; SERCA1 levels in WT mice.). Moreover, please show the red ponceau or another loading control as GAPDH is a metabolic enzyme, the expression of which could be different in the various mouse models.
Minor comments.
1) Please indicate the supplier reference of the antibody used.
2) The results are not always depicted as mean +/- SEM. Please correct section 2.6 and the figure legends.
3) Please indicate in section 2.6, or for each figure as done for figure 3 the post-hoc analysis used.
4) The legend of figure 1 is incorrect.
5) Please verify the sentence line 209.
6) line 221. The body fat was estimated and not calculated.
7) line 231. The sentence is unclear.
8) line 288-291. Please remove this paragraph as it is repeated in the discussion (line 373).
Round 2
Reviewer 2 Report
The authors pointed out that their expertise is skeletal muscle and underlined that their findings open new avenue for the study of the adipose tissue. As the time given for the revision does not allow to repeat the immunoblottings, they replied to most of the various points raised.